# The impact of model assumptions in interpreting cell kinetic studies

Ada Wing Chi Yan[1], Ildar Sadreev[1], Jonas Mackerodt[1], Yan Zhang[2], Derek Macallan[2], Robert Busch[3], Becca Asquith[1]*

**1** Department of Infectious Disease, Imperial College London, London, United Kingdom, **2** Institute for Infection and Immunity, St George's, University of London, London, United Kingdom, **3** Department of Life Sciences, University of Roehampton, London, United Kingdom

* b.asquith@imperial.ac.uk

**Data availability statement:** Code to reproduce results can be found at https: //github.com/ada-w-yan/cellkineticmodels. Only

## Abstract

Stable isotope labelling is one of the best methods currently available for quantifying cell dynamics *in vivo*, particularly in humans where the absence of toxicity makes it preferable over other techniques such as CFSE or BrdU. Interpretation of stable isotope labelling data (as for BrdU and CFSE) necessitates simplifying assumptions. Here we investigate the impact of three of the most commonly used simplifying assumptions: (i) that the cell population of interest is closed, (ii) that the population of interest is kinetically homogeneous, and (iii) that the population is spatially homogeneous and suggest pragmatic ways in which the resulting errors can be reduced.

## Author summary

Our immune response is responsible for protecting us from infectious disease and cancer. The immune system is not static, immune cells are consistent proliferating and dying even in healthy individuals. Understanding the rate at which immune cells proliferate and die is an important to understanding how our immune systems function; for example how immune memory is maintained or how a diverse repertoire of cells capable of responding to any invading pathogen is generated. Cell dynamics are usually measured using a tracer dye. A method called stable isotope labelling is one of the best methods for quantifying cell dynamics in humans *in vivo* as it is nontoxic. However, interpretation of any labelling experiment requires assumptions. In this paper we discuss the impact of some commonly used assumptions and suggest ways in which the resulting errors can be reduced.

## Introduction

The quantification of cell dynamics is a key component of our understanding of human physiology in health and disease. Stable isotope labelling is considered the gold standard for the measurement of cell dynamics in humans *in vivo*. Stable isotope labelling has been used to

simulated data was used in the study; full details of how to reproduce these data are given in the methods.

**Funding:** AY is funded by an Imperial College Research Fellowship. BA is a Wellcome Trust (WT) Investigator (103865Z/14/Z) and is also funded by the Medical Research Council (MRC) (J007439, G1001052), the European Union Seventh Framework Programme (FP7/2007–2013) under grant agreement 317040 (QuanTI), the European Union H2020 programme under grant agreement 764698 (QUANTII) and Leukemia and Lymphoma Research (15012). The funders had no role in study design, data collection and analysis, decision to publish, or preparation of the manuscript.

**Competing interests:** The authors have declared that no competing interests exist.

investigate the dynamics of a wide range of cells, particularly haematopoietic cells, including neutrophils, monocytes and lymphocytes [1–6] and has led to fundamental discoveries in both basic cell biology [7–11] and in the pathology of diseases including HIV-1, chronic lymphocytic leukemia and type 1 diabetes [12–15].

In a typical stable isotope labelling experiment, volunteers will be given a stable isotope (usually deuterium in the form of deuterated water or deuterated glucose) for a fixed period of time: the labelling period. Blood will be drawn at multiple timepoints, during the labelling and delabelling phase, the cells of interest will be sorted by flow cytometry, DNA extracted and the fraction of deuterium-labelled nucleotides quantified by gas chromatography/ mass spectrometry. Available deuterium is incorporated when DNA is synthesised and lost when labelled cells disappear (die, change phenotype or exit the sampled compartment long-term). In this respect, stable isotope labelling is simpler to interpret than techniques that count labelled cells (as is common with BrdU) since, when relying on the count of labelled cells, division of a labelled cell can result either in the production of an extra labelled cell (if both daughter cells still have high enough concentration of BrdU to count as BrdU-positive) or the loss of the labelled cell (if division and therefore dilution of the label means both daughter cells now fall below the threshold of "BrdU-positive"); i.e. upon division one BrdU labelled cell can become either 2 or zero labelled cells. For stable isotope labelling, the fraction of labelled nucleotides over time contains information about cell proliferation and cell disappearance. To extract this information mathematical models are constructed and fitted to the data [16].

Mathematical models for interpreting stable isotope labelling data are necessarily simple. Over-complicated models would result in unidentifiable parameters and would not be fit for the purpose of parameter estimation. Whilst essential, these model simplifications do, potentially, impact on parameter estimates. Here we explore the impact of three widely used simplifying assumptions and suggest pragmatic ways in which the resulting errors in parameter estimates can be reduced.

The first assumption we investigate is that the cell population of interest is closed i.e. there are no upstream or downstream compartments feeding into or out of the target compartment of interest. The second assumption is that the population of interest is kinetically homogeneous i.e. all cells in the population have the same kinetics and the third is that the population is spatially homogeneous i.e. a sample of a population from the blood would yield the same kinetics as a sample of the same population from a spatially-distinct tissue.

## Results

### 1. Upstream and downstream compartments

Cell populations are not isolated, closed systems: cells enter a target population as they mature and change phenotype (to the phenotype of the target population) and leave the population as they die, exit the sampled compartment long-term or change phenotype (from the phenotype of the target population). Labelling data will often only be collected for the target cell population of interest and not for compartments upstream and downstream of the target cell compartment. There are multiple reasons why this may be the case: sometimes the lineage topology is unknown and so the identity of the upstream and downstream compartments is unknown, sometimes it is because these compartments are difficult to access in humans (e.g. maturation occurs in the bone marrow or thymus) and sometimes because blood volume is limited and so populations other than the target population cannot be sorted. We have previously shown that having an upstream compartment can affect the fraction of measured label, and potentially lead to incorrect estimation of the kinetic parameters [1,17]. Having a

downstream compartment does not affect parameter estimation, provided that the disappearance rate is interpreted as the loss of cells due to all causes including exit to the downstream compartment.

Here, we investigate the conditions under which the kinetic parameters of the target cell compartment can be estimated accurately despite assuming that the target compartment is closed when model fitting. We also investigate whether fitting a model with an upstream compartment yields more accurate and/or precise estimates of the kinetic parameters compared to fitting a single-compartment model in the event that data from the upstream compartment is absent.

We consider a scenario (Fig 1) in which the target population of interest, $E$, is sampled but its upstream, precursor compartment, $C$, is not. Upstream, precursor cells $C$ proliferate at rate $p_C$, die at rate $d_C$, and differentiate at rate $r$. Differentiation is allowed, but not required, to be linked to clonal expansion. Each precursor cell divides $k$ times before differentiation (the case $k=0$ corresponds to differentiation in the absence of division). Clonal expansion is assumed to occur outside of the sampled compartment (typically blood). Target population cells, $E$, proliferate at rate $p_E$ and disappear (die or differentiate) at rate $d_E$. The number of cells in the two populations is assumed to be independently at equilibrium. Then the dynamics of the two cell populations follow the equations

$$\frac{dC}{dt} = p_C C - d_C C - rC \tag{1a}$$

$$\frac{dE}{dt} = r2^k C + p_E E - d_E E. \tag{1b}$$

Upon stable isotope labelling (discussed here without loss of generality for labelling with deuterated water), then the fraction of label in each compartment ($F_C$ and $F_E$ respectively) is

$$\frac{dF_C}{dt} = p_C b_w U(t) - (d_C + r)F_C, \tag{2a}$$

$$\frac{dF_E}{dt} = (2^k - 1)b_w U(t)r\frac{\bar{C}}{\bar{E}} + r\frac{\bar{C}}{\bar{E}}F_C + p_E b_w U(t) - d_E F_E \tag{2b}$$

where $\bar{C}$ and $\bar{E}$ are the equilibrium population sizes for $C$ and $E$, $b_w$ is the normalisation factor for water [18] and $U(t)$ is the fraction of label in body water.

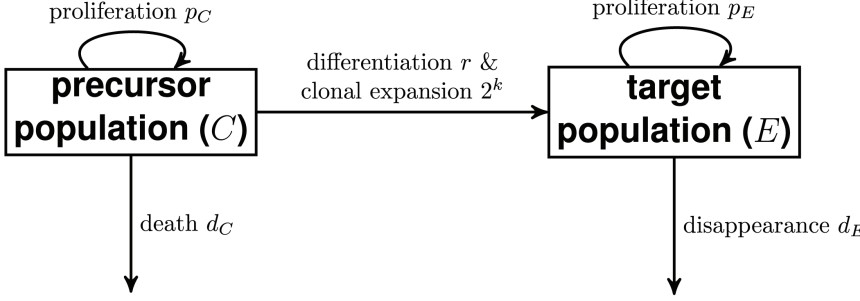

Fig 1. Compartmental diagram for model with precursor (upstream) and target populations.

We describe the fraction of deuterium in body water using a simple empirical curve

$$U(t) = \begin{cases} f[1 - \exp(-\delta t)] & \text{if } t \le \tau \\ U(\tau)\exp(-\delta(t-\tau)) & \text{otherwise} \end{cases} \quad (3)$$

where, $f$ is the plateau enrichment (expected to be the fraction of D$_2$O in the daily intake of water) and $\delta$ the rate at which the plateau is approached (expected to be equivalent to the turnover rate of body water) and $\tau$ is the length of the labelling phase.

If each compartment ($C$, $E$) is kinetically homogeneous – that is, all cells in the compartment have the same kinetics rate – then in Eqs 1 and Eqs 2, equilibrium constraints require $p_C = d_C + r$ and $r2^k \frac{\bar{C}}{\bar{E}} + p_E = d_E$. Kinetic heterogeneity in each compartment, where the precursor and/or target populations are composed of subpopulations with different turnover rates, can be approximated by replacing $d_C$ and $d_E$ in Eq 2 with $d_C^*$ and $d_E^*$ respectively where $d_C^*$ and $d_E^*$ are the disappearance rates of the labelled cells in compartments $C$ and $E$ respectively, and we typically expect $d_C^* \ge d_C$, $d_E^* \ge d_E$ [19].

Focusing on the target compartment (our cell population of interest), we describe its kinetics in terms of three descriptors:

i  its proliferation rate, defined to be $p_E$
ii  its turnover rate, defined to be the total rate of inflow into the target compartment, $T = r2^k \bar{C}/\bar{E} + p_E$, which is equal to the rate of outflow (disappearance rate across all cells), $d_E$, at steady state
iii  its rate of production by division defined to be $P = r(2^k - 1)\bar{C}/\bar{E} + p_E$.

Fig 2 shows the relationships between target compartment parameters for four cases. In case A, $k>0$ and the influx of labelled cells from C to E is large. In case C, $k = 0$ and the influx of unlabelled cells from C to E is large. In cases B and D, the influx of cells from C to E is small; $k>0$ and $k = 0$ respectively. The ratio between the influx of cells from C to E and the total turnover can be written as $X = \frac{r2^k \bar{C}}{d_E \bar{E}}$. In cases A and C, this is significantly larger than 0 (but smaller than 1 by definition); in cases B and D, it is close to 0.

**Fitting a one-compartment model to the target data**  When we only have measurements of $F_E$ (i.e. label in the target population), one commonly used option is to neglect the existence of the precursor population and to fit a one-compartment model. In this study we will focus on fitting the version of the model with implicit kinetic heterogeneity, as it is a more general version of the homogeneous model but retains parameter identifiability (unlike models which explicitly describe multiple sub-populations within the target population, see

| | | Proliferation $p_E$ | | Production by division $r(2^k-1)\dfrac{\bar{C}}{\bar{E}}+p_E$ | | Turnover (= disappearance of all cells) $r2^k\dfrac{\bar{C}}{\bar{E}}+p_E$ | | Disappearance of labelled cells $d_E^*$ |
|---|---|---|---|---|---|---|---|---|
| (A) | $k > 0$ , large influx of labelled cells from C to E | | $\ll$ | | $\approx$ | | $\le$ | |
| (B) | $k > 0$ , small influx of cells from C to E | | $\approx$ | | $\approx$ | | $\le$ | |
| (C) | $k = 0$ , large influx of unlabelled cells from C to E | | $=$ | | $\ll$ | | $\le$ | |
| (D) | $k = 0$ , small influx of cells from C to E | | $=$ | | $\approx$ | | $\le$ | |

**Fig 2. Relationships between target compartment parameters for four cases.**

next section):

$$\frac{dF}{dt} = pb_wU(t) - d^*F. \tag{4}$$

In Eq 4, proliferation, production by division and turnover are described by a single rate $p$. This will evidently cause problems when fitting to the target data, as Fig 2 shows that these terms are not equal when the influx of cells into the target compartment is large. In S1 File, we show that the estimated $p$ (which we will refer to as $\hat{p}$) most closely approximates the production rate by division, which for cases A, B and D is approximately equal to the turnover rate, and for cases B, C and D is approximately equal to the proliferation rate. Thus for case A when there is a large influx of labelled cells into the target compartment, the proliferation rate is underestimated by $\hat{p}$, and for case C when there is a large influx of unlabelled cells into the target compartment, the turnover rate is overestimated by $\hat{p}$. As the error occurs due to the conflation of the proliferation rate, production rate by division and turnover rates, its magnitude is equal to that of the true discrepancy between these rates in the target compartment, which is not possible to assess with information from the target compartment only.

If we know the identity of the upstream compartment, we may have some information about $p_C$ (its proliferation rate) and $\frac{\bar{C}}{\bar{E}}$ (the ratio between the two population sizes) from the literature. As $p_C$ is an upper bound for $r$, $p_C\frac{\bar{C}}{\bar{E}}$ is an upper bound for the difference between the rates of turnover and production by division. However, without knowledge of $k$, it is difficult to find a bound for the difference between the rates or proliferation and production by division.

We demonstrate these results with simulated data. S2 File shows 100 parameter sets sampled from the prior distribution (Table 1, S1 Fig "optimal data"), and Figs A-D in S3 File show the target compartment labelling data simulated using the two compartment precursor-target model and these parameters. Fig 3 shows the results of then fitting the one compartment kinetic heterogeneity model (Eq 4) to these data (Methods). $\hat{p}$ and $\hat{d}^*$ are the estimated values of $p$ and $d^*$, respectively. The top row shows results for $1 \leq k \leq 20$ (cases A and B in Fig 2). As expected, $\hat{p}$ is a good estimate of the turnover rate of the target compartment $r2^k\bar{C}/\bar{E} + p_E$ and of the rate of production of target cells by division $r(2^k - 1)\bar{C}/\bar{E} + p_E$. However, it is a poor estimate of the true proliferation rate of the target compartment $p_E$. The bottom row shows results for $k = 0$ (cases C and D in Fig 2). Turnover is now underestimated by $\hat{p}$ but the proliferation rate and the rate of production by cell division are accurately captured.

Fig 3D and H show that if $k = 0$, estimates of $d^*$ are potentially inaccurate, i.e. differ from the true disappearance rate of labelled cells. This observation is further analysed in S1 File. This is arguably less problematic as $d^*$ is not a stable, intrinsic descriptor of the population as it depends on the labelling protocol.

In Fig 4, we break down the errors by influx ratio in addition to $k$. Fig 4A shows the error in the production rate by division when fitting a one-compartment model to the simulated data sets, defined as a percentage $100\frac{\hat{p}-P}{P}$. As expected from the theoretical results in S1 File, the error in the production rate by division is small, unless $k = 0$ and the influx ratio is large (case C in Fig 2). In this case, the $F_C$ term in $\frac{dF_E}{dt}$ in Eq 2 is relatively large, leading to error in estimating $P$. In Fig 4B, for each value of the influx ratio $X$ on the x-axis and $k$ indicated by the colour, the lines show the theoretical error from assuming that the production rate by division is equal to the proliferation rate, i.e. $100\frac{P-p_E}{p_E}$. This error can be rearranged into $100\frac{X(2^k-1)}{(1-X)2^k}$, so it is a function of $X$ and $k$ only. We can see, for example, that if an error of less than 10% is needed, this can be achieved with an influx ratio $X < 0.1$ regardless of $k$; but the requirement becomes less stringent for $k = 0$. Hence, proliferation is well approximated by the production rate by division for cases B, C and D in Fig 2. The dots show $100\frac{\hat{p}-p_E}{p_E}$ estimated

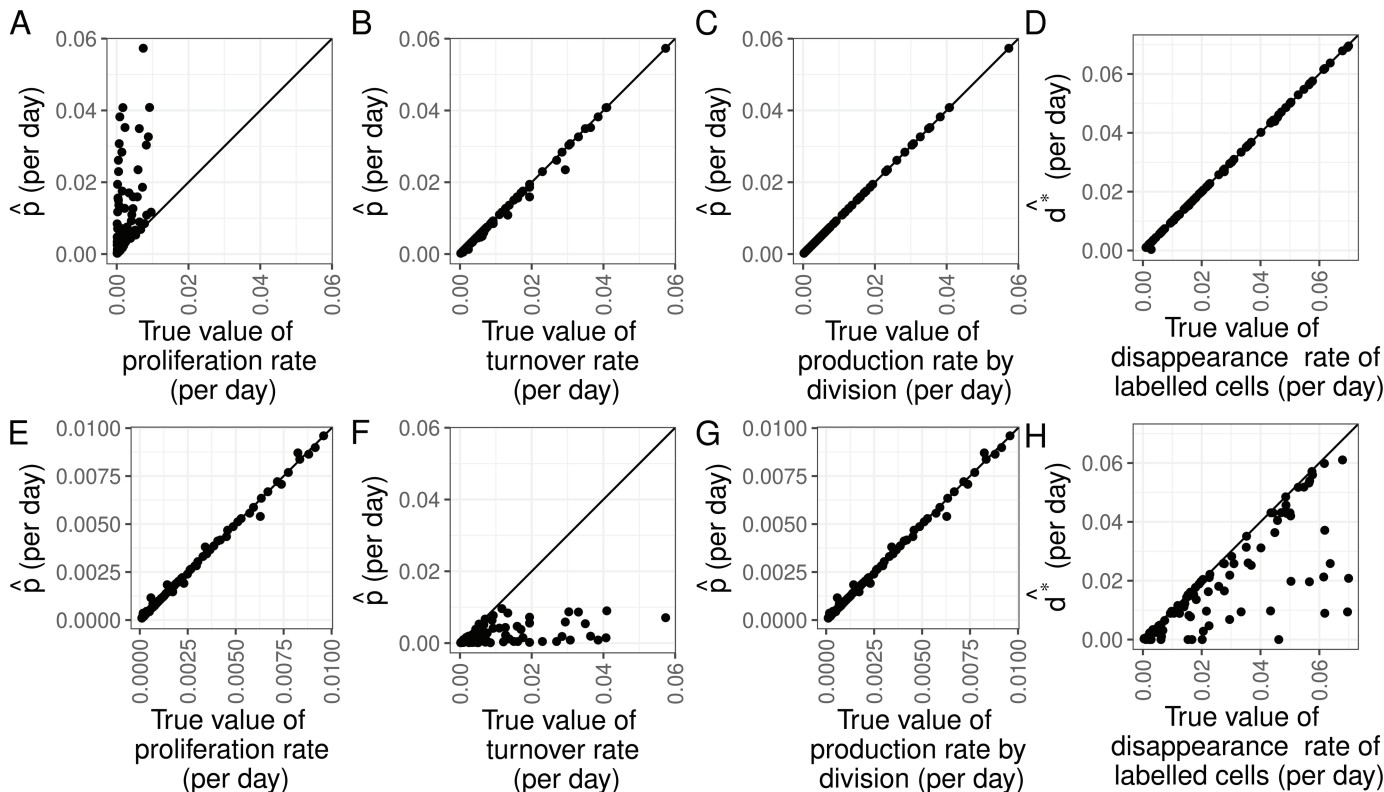

**Fig 3. Estimated $\hat{p}$ (all columns except rightmost column) and $\hat{d}^*$ (right column) compared to (from left to right) the rates of proliferation, turnover, production by division and the disappearance rate of labelled cells in the target compartment.** Top: for the case when $k$ (the number of rounds of cell division associated with differentiation) varies in the range [1,20]; bottom: when $k$ is constrained to zero.

from the data. For $k > 0$, the estimated errors agree well with the theoretical results. For $k = 0$, the theoretical result is zero error, but in reality there is still some error due to the error in the estimation of $p$ as shown in Fig 4A. In Fig 4C, the lines show the theoretical error from assuming that the production rate by division is equal to the turnover rate, $T$, i.e. $100\frac{|P-T|}{T}$. This error can be rearranged into $\frac{X}{2^k}$, so it is a function of $X$ and $k$ only. It increases with $X$ and decreases with $k$; for example, if an error of less than 25% is needed, this can be achieved with $k \geq 2$ for all values of $X$, or $X \leq 0.5$ for $k = 1$, or $X \leq 0.25$ for $k = 0$. The dots show $100\frac{\hat{p}-T}{T}$ estimated from the data, which agree with the theoretical results. Hence, turnover is well approximated by the production rate by division for cases A, B and D in Fig 2.

In a companion article by Swain et al. [35], the special cases $k = 0$ and $k = 1$ with kinetic homogeneity ($d_C^* = d_C$, $d_E^* = d_E$) are examined. These studies agree with us that for $k = 0$, turnover is underestimated by $\hat{p}$, unless the precursor cells turn over quickly.

We conclude that, when there is a significant flow of labelled cells into the target population of interest ($k \gg 0$, precursor population very large or proliferating rapidly), then we (1) cannot estimate proliferation of the target population but (2) can calculate an upper bound on the proliferation rate and (3) can also estimate target cell production by division which is approximately equal to the turnover. The approximation between turnover and production by division holds provided there is division linked differentiation. The larger the value of $k$ (the number of cell divisions upon differentiation) the better this approximation. Conversely, for low values of $k$, the influx of cells into the target population from the upstream population

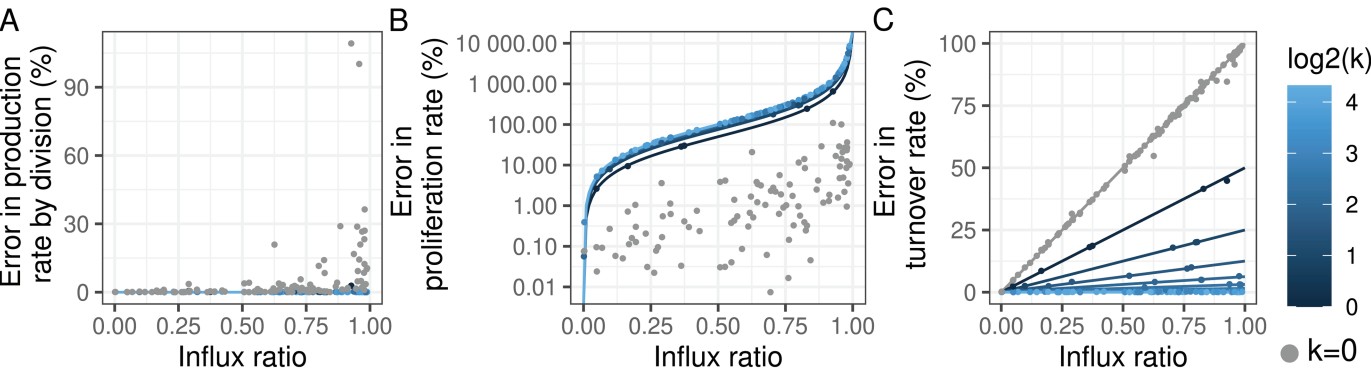

**Fig 4. Errors in estimating the rates of production by division, proliferation and turnover, as a function of the influx ratio $X$ and the number of divisions linked to differentiation $k$.** (A) The error in the production rate by division when fitting a one-compartment model to the simulated data sets, defined as a percentage $100\frac{\hat{p}-P}{P}$. (B) For each value of the influx ratio $X$ on the x-axis and $k$ indicated by the colour, the lines show the theoretical error from assuming that the production rate by division is equal to the proliferation rate, i.e. $100\frac{P-p_E}{p_E}$. The dots show $100\frac{\hat{p}-p_E}{p_E}$ estimated from the data. (C) Same as (B) but for the turnover rate.

will be unlabelled and therefore invisible to the experiment. In this scenario, we can (1) estimate the proliferation rate (2) estimate the production by division (3) obtain an upper bound on the turnover rate. It is straightforward to understand these results intuitively. When fitting a one compartment model, label from any source, be it proliferation of the target population, proliferation of the precursor or clonal division upon differentiation is subsumed into one term (i.e. there is no way to distinguish between these sources of label) and quantified by the parameter $p$. The correct interpretation of this is label that has arisen by division which we refer to as the production by division. $\hat{p}$ is therefore in all cases an accurate estimate of the production by division. In the past, $\hat{p}$ has been incorrectly interpreted as the proliferation of the target population. The total label arising from division is the sum of the label arising from the target population proliferation and the label arising from the upstream population; therefore, in the case where there is little label from the upstream population ($k = 0$), then the production by division is approximately equal to the proliferation and $\hat{p}$ approximates the proliferation rate. This approximation breaks down as the amount of label from the upstream population increases. On the other hand, turnover is equal to the gain (or loss) of all cells by the target population. Flow of unlabelled cells from the upstream population will be invisible in the labelling experiment, so if this flow is relatively large (e.g. the $k = 0$ case), then turnover will be underestimated by the production by division. However, if this term is small (e.g. $k \gg 0$) then production by division will be similar to the turnover and therefore turnover will be accurately approximated by $\hat{p}$. These relationships are summarised in Fig 5.

**Comparing fits using models with and without an upstream compartment** Next we investigate whether, if proliferation rather than turnover is the quantity of interest, and clonal expansion contributes significantly to label ($k \gg 0$), there is any benefit in fitting the two compartment precursor/target model (Eq 2) despite an absence of data from the upstream precursor compartment. That is, we consider the same simulated data as in the previous section, but fit the precursor/target model. When performing Bayesian inference, the chains do not converge so we utilised a maximum likelihood approach instead. This allows us to obtain point estimates $\hat{p}_E$ and $\hat{d}_E^*$, but fit diagnostics indicate that parameters are not identifiable due to colinearity and the covariance matrix is singular. Although the diagnostics indicated that the fits are unreliable, the point estimates of the proliferation rate from fitting the two compartment

| | | Target population descriptor | | | |
|---|---|---|---|---|---|
| | | **Production by division** | **Turnover** | **Proliferation** | **Disappearance of labelled cells** |
| **Fit the one compartment model** | **Upstream compartment contributes relatively large number of unlabelled cells e.g. k=0** | accurately estimated by $\hat{p}$ | $\hat{p}$ is a lower bound | accurately estimated by $\hat{p}$ | often accurately estimated by $\hat{d}^*$ |
| | **Upstream compartment contributes relatively large number of labelled cells e.g. k>>0** | accurately estimated by $\hat{p}$ | accurately estimated by $\hat{p}$ | $\hat{p}$ is an upper bound | accurately estimated by $\hat{d}^*$ |
| | **Upstream compartment contributes relatively few cells** | accurately estimated by $\hat{p}$ | accurately estimated by $\hat{p}$ | accurately estimated by $\hat{p}$ | accurately estimated by $\hat{d}^*$ |

**Fig 5. Summary of errors in estimation of target population descriptors when fitting a one-compartment model to data from the target compartment.**

precursor/target model are closer to the true values than when fitting the one compartment model.

The errors in the proliferation rate estimates are significantly lower when fitting the precursor/target model than when fitting the one population model (median 83% and 201% respectively; $P = 2.10^{-9}$, $N = 98$, Wilcoxon signed rank), Fig 6A. We reason that in an experimental setting where the identity of the precursor compartment is known but not sampled then the ratio of sizes of the precursor and target compartments can be added as a fixed parameter and that this might aid parameter estimation; we refer to this as the precursor/target model with ratio. Unexpectedly, providing the ratio of the size of the upstream to target compartments does not result in a reduction in error. If anything the errors in parameter estimates from the precursor/target model with ratio are slightly higher than the estimates from the precursor/target model without the ratio (median 109% and 83% respectively; $P = 0.04$, Wilcoxon signed rank). However, errors still remain significantly lower than for the one compartment model, Fig 6A.

For each of the three models, for the 100 data sets we also estimate the accuracy, i.e. what proportion of times the true parameter values lay within the 95% confidence intervals (CI) of the estimated values (calculated by bootstrapping the data, Methods), Fig 6B. For all three models, the proportion of runs where the true parameter value lies within the 95% CI of the estimate is low (3.6%, 8.3% and 27% for the one compartment model, precursor/target model and precursor/target with ratio model respectively). The proportion is significantly higher for the precursor/target with ratio model than for either the one compartment model ($P = 6 \times 10^{-5}$, Fisher's exact test) or the precursor/target model ($P = 0.003$, Fisher's exact test). The low proportion of runs where the estimate lies within the CI indicates that for all three models, for the data set in question, estimating the CI by bootstrapping the data leads to a considerable underestimate of the CI; the asymptotic covariance matrix method on the other hand finds infinite intervals.

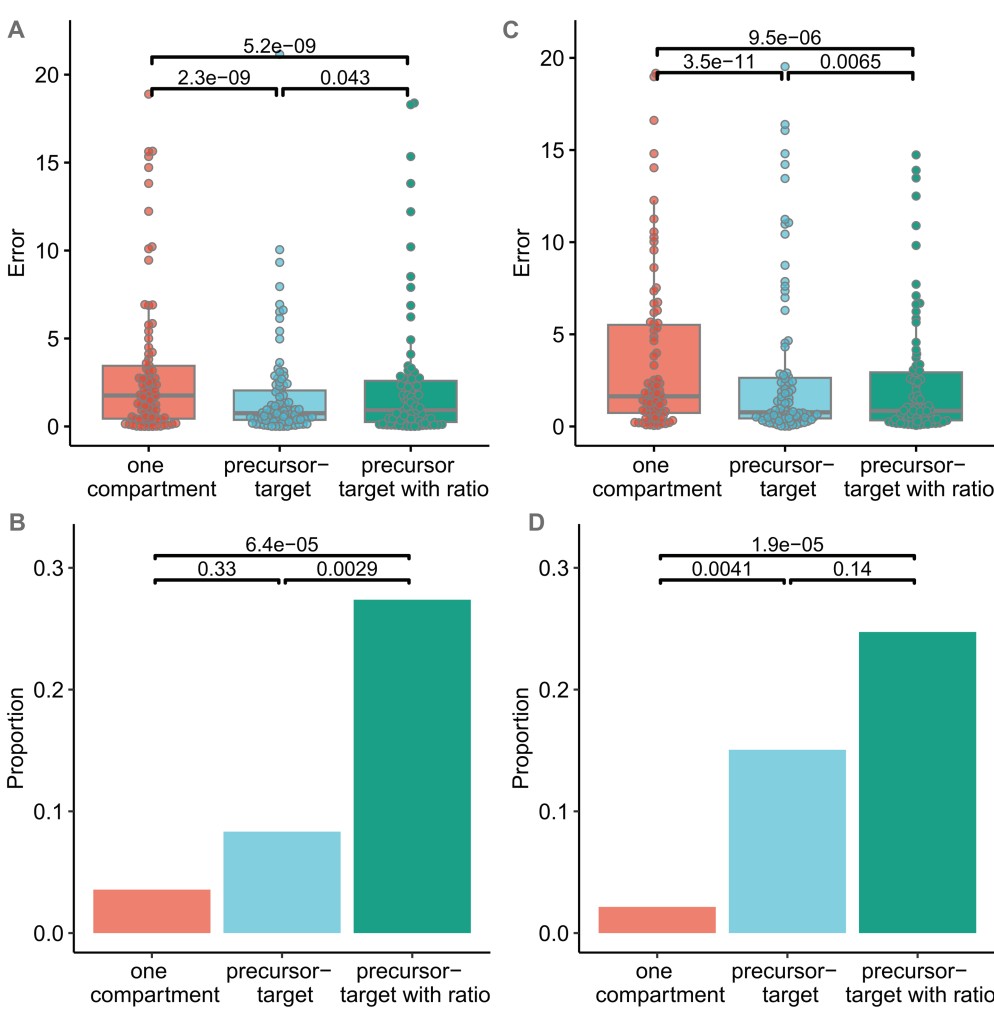

**Fig 6. Error in estimates of the proliferation rate. A** Discrepancy between point estimate of the proliferation rate and the true value for optimal data (expressed as a proportion of the true value) for the one compartment target model (red), precursor/target model (blue) and precursor/target with ratio model (green). Note the y axis has been truncated to show the bulk of the data more clearly; the same graph but with no y axis truncation is given in S3 Fig. P values are calculated by Wilcoxon signed rank, two-tailed; not corrected for multiple comparisons (number of independent comparisons ≤ 3). **B** 95% confidence intervals (CI) were estimated by bootstrapping the data (Methods) and the fraction of runs where the true value lay within the CI is reported. Colours as for A. P values are calculated by Fisher's Exact Test; not corrected for multiple comparisons (number of independent comparisons ≤ 3). The analysis in A and B was then repeated for more realistic data in **C** and **D**.

When fitting the precursor/target model, $\hat{p}_E$ now only corresponds to proliferation and not production by division or turnover, so as expected, the latter quantities are poorly estimated by $\hat{p}_E$ (5S Fig).

To this point we have been fitting optimal data in order to have a simpler system to aid analytical work and to focus on the effect of an upstream compartment rather than on imperfect data. There are three main respects in which our simulated data so far has been optimal: 1) noise is low 2) sampling is frequent 3) heterogeneity in the target compartment is represented by a single exponential with $d^* > p$ rather than explicitly with subpopulations). We therefore repeat the analysis using more realistic data by relaxing these three constraints

(Methods). The parameters for model simulation are in S2 File, and the simulated data sets are in Figs E-H of S3 File. On calculating the errors for all three models fitted to this more realistic data we find the same pattern: i.e. the errors in the estimates of the proliferation rate obtained by fitting the precursor/target model are significantly lower than those obtained by fitting either the one compartment model or the precursor/target model with ratio (Fig 6C). Additionally, as for the optimal data, we also find that the highest proportion of runs where the true value of the parameter lies within the 95% CI of the estimated value is seen for the precursor/target model with ratio (Fig 6D). However, when changing the parameter ranges with which the simulated data are produced so that the mean proliferation of the target population is $p_E = 0.02$ (compared to a mean of $p_E = 0.002$ for the previous data sets) then relative errors produced by all models are much smaller and there are no longer any significant differences in the errors between the models (S4 Fig). We find then that, at least for values of proliferation that are typically seen with T cells (i.e. the results shown in the main text rather than S4 Fig), fitting a precursor/target model, even when only target cell data is available produces estimates of the target cell proliferation rate with a lower percentage error.

In summary, we recommend that where it is suspected that there may be an upstream compartment but this compartment has not been sampled then the one compartment implicit kinetic heterogeneity model (Eq 4) is fitted and the estimated parameter $\hat{p}$ is interpreted as the production rate by division. This interpretation will always be correct independent of the dynamics or relative size of the upstream compartment. If $\hat{p}$ is equated with proliferation and/or turnover then the necessary assumptions should be spelt out (that there are no labelled cells entering from an upstream source and that there are no unlabelled cells entering from an upstream source respectively). Ideally, further information should be obtained about the size and dynamics of the upstream population. In the case where there is considerable label entering from the upstream compartment and we wish to quantify how much of the production by division is attributable to the proliferation of the target population then an improved estimate of the proliferation can be made by fitting the precursor/target model though this may suffer from large errors.

## 2. Kinetic heterogeneity

The second assumption we consider is the assumption of kinetic homogeneity. When estimating rates of cell turnover, a common assumption is that the population is kinetically homogeneous, i.e. all cells of the population proliferate and disappear at the same rate. However, the population may actually be composed of subpopulations which each have different kinetics, which is termed kinetic heterogeneity in the literature [19,20].

Kinetic heterogeneity can be modelled explicitly. If we assume the simplest form of kinetic heterogenity in which there are $N$ unconnected subpopulations then the dynamics of the subpopulations obey

$$\frac{dC_i}{dt} = p_i C_i - d_i C_i, \qquad i = 1, 2, ..., N \tag{5}$$

If each subpopulation is independently at equilibrium, $p_i = d_i$, and we can define the relative size of each population as $\alpha_i = \frac{C_i}{\sum_{j=1}^{N} C_j}$. This model is illustrated for two subpopulations in Fig 7.

The fraction of labelled cells in this explicit kinetic heterogeneity model is [20]

$$\frac{dF_i}{dt} = p_i b_w U(t) - d_i F_i, \qquad i = 1, 2, ..., N, \tag{6a}$$

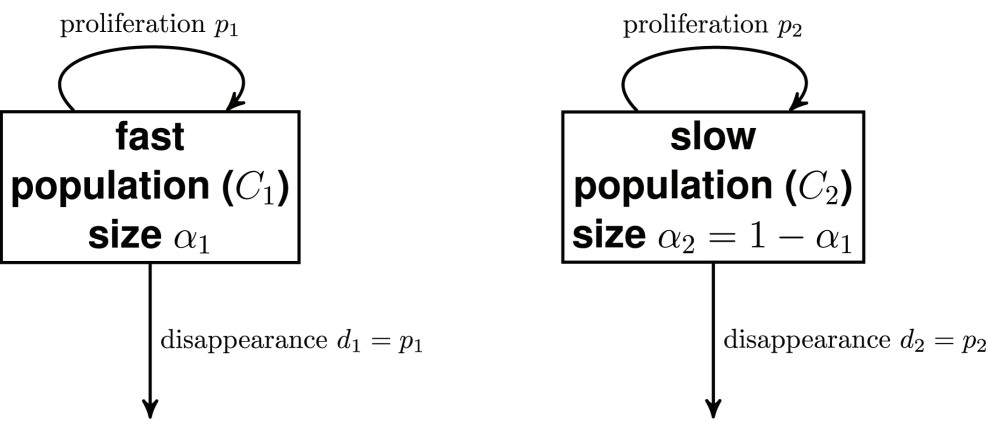

**Fig 7. Compartmental diagram for two-compartment model with explicit kinetic heterogeneity.**

$$F = \sum_{i=1}^{N} \alpha_i F_i. \tag{6b}$$

Here, $F_i$ is the fraction of labelled cells in subpopulation $i$ and $F$ is the fraction of labelled cells in the total population. The companion article by Swain et al. [35] notes that the target compartment labelling curve in the precursor/target model is identical in form to the labelling curve for the two subpopulations case if the target cells turn over faster than the precursor population. In this case it is not possible to distinguish between the two possible sets of dynamics underlying the labelling curve.

In Eq 6 the number of parameters scales linearly with the number of subpopulations and moreover there are strong correlations between the parameters, so some parameters may not be identifiable with limited data. A simplification can be made to yield the implicit kinetic heterogeneity model Eq 7 [19], where $F$ is the overall fraction of labelled cells, $p$ is the (weighted) mean proliferation rate across subpopulations and $d^*$ is the disappearance rate of labelled cells, which is biased towards cells with the fastest turnover.

$$\frac{dF}{dt} = p b_w U(t) - d^* F \tag{7}$$

Previous studies have addressed the implications of kinetic heterogeneity on parameter estimation [6,21–23]. For a cell population consisting of subpopulations individually in equilibrium then the explicit kinetic heterogeneity model encapsulates the true dynamics of the system. By reducing the number of model parameters, the implicit kinetic heterogeneity model represents an approximation that potentially results in inaccurate estimates. This was shown by Westera et al. [23]. They further showed that the estimation of average proliferation could be improved by fitting an explicit heterogeneity model with multiple, explicitly modelled, subcompartments and demonstrated this by comparing parameter estimates obtained by fitting this model [23]. However, this study used a single parameter set; we now explore the generalisability of the results.

**Implicit kinetic heterogeneity model parameter estimates are more precise.** We simulated data using a two-compartment explicit model, for baseline parameters given in Table 2, varying $p_1$, $p_2$ and $\alpha_1$ (Methods, 125 simulations in total). We then fitted the homogeneous,

implicit heterogeneous and explicit heterogeneous (two-compartment) models to the simulated data. Fig 8A shows the percentage error (as previously defined) in estimates of the mean proliferation rate for each of the three models ("model selection" will be discussed later in the manuscript). Surprisingly, the errors tended to be larger for the explicit model compared to the implicit model (p = 0.0047, Wilcoxon signed rank test), inconsistent with the results of Westera et al. [23]. However, the explicit model does have a higher proportion of parameter sets for which the 95% CI for the mean proliferation rate contains the true value (Fig 8B), as the width of the CIs tends to be larger (Fig 8C). Parameter estimates for individual data sets are shown in S6–S11 Figs.

We investigated how the performance of each model depended on the parameters used to generate the simulated data. Fig 9 shows that when $p_1 \leq 10p_2$, the implicit kinetic heterogeneity model has a low percentage error regardless of the other parameter values, while when $p_1 > 10p_2$, the errors for the implicit kinetic heterogeneity model are large when $\alpha_1$ is small, but decrease when $\alpha_1$ is large. Westera et al. [23] simulated using the parameters $p_1 = 0.72$, $p_2 = 0.016$ and $\alpha_1 = 0.1$, which falls in the region of parameter space ($p_1 \gg p_2$ and $\alpha_1$ small) where the estimate of the mean proliferation rate from the implicit kinetic heterogeneity model has a large error.

This result illustrates a fundamental problem with estimating the mean proliferation rate using stable isotope labelling: that it is difficult to rule out the existence of a fast, small compartment which, while not changing the overall proportion of labelled cells much, does drive up the mean proliferation rate. S12 Fig shows pairs plots of the posterior distribution for fitting the two-compartment model to data generated with two different sets of parameter values; both show this lack of identifiability for $p_1$ and $\alpha_1$.

The implicit kinetic heterogeneity model uses a single parameter for the disappearance rate of labelled cells, even though the proportion of labelled cells with a fast turnover decreases during the delabelling period [19]. Thus, it encodes an implicit assumption that the amount of kinetic heterogeneity is small, which narrows down the posterior distribution of the mean proliferation rate compared to the explicit model. When the true amount of kinetic heterogeneity is indeed small, then the implicit kinetic heterogeneity model thus produces both

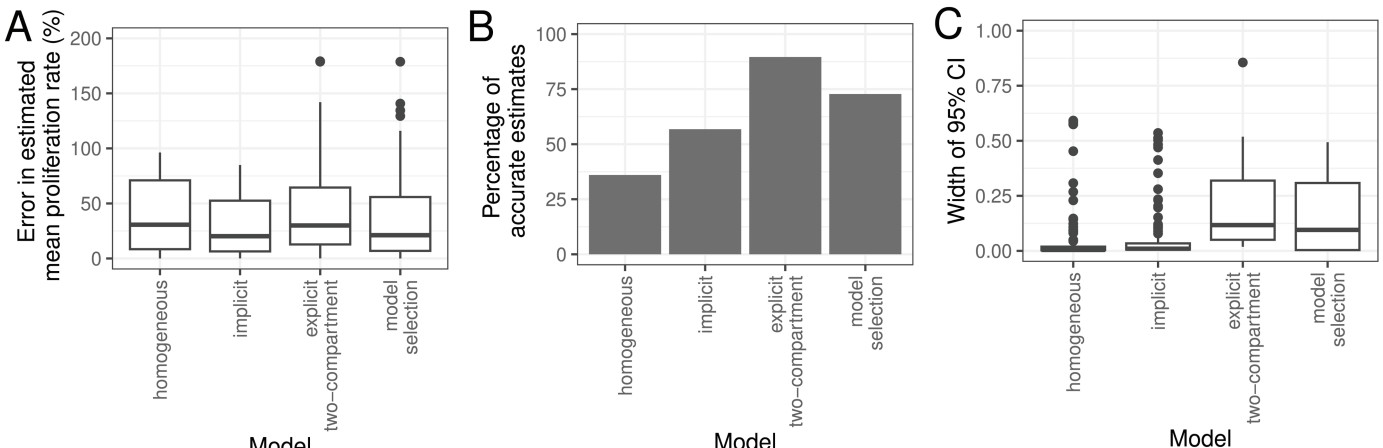

**Fig 8. Evaluating model fits for different models of kinetic heterogeneity.** (A) The percentage error in the mean proliferation rate, (B) the percentage of simulations for which the true value of the mean proliferation rate lies within the 95% CI, and (C) the width of the 95% CI of the mean proliferation rate, across 125 simulations of the two-compartment model, when the models on the x-axis are fitted to the data.

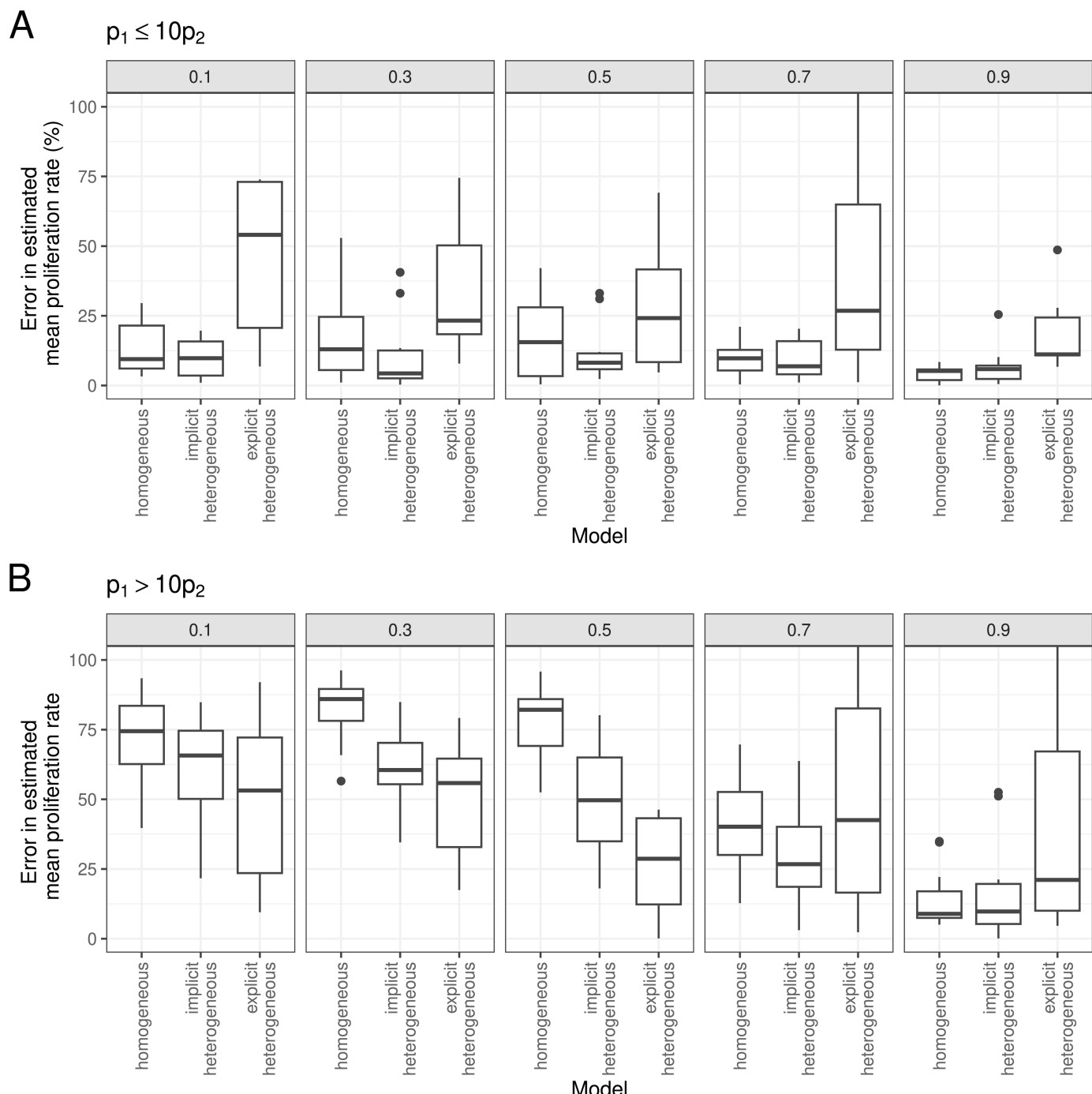

**Fig 9. Evaluating model fits for different models of kinetic heterogeneity, stratified by the ratio between $p_1$ and $p_2$.** (A) The percentage error in the mean proliferation rate for (A) $p_1 \leq 10p_2$ and (B) $p_1 > 10p_2$, for different values of $\alpha_1$ (subplots), across 125 simulations of the two-compartment model, when the models on the x-axis are fitted to the data.

more precise and accurate estimates; however, when the true amount of kinetic heterogeneity is large, the implicit kinetic heterogeneity model can be inaccurate.

**Explicit kinetic heterogeneity model parameter estimates are very sensitive to the choice of prior distributions.** Although the explicit two-compartment model had a higher proportion of simulations where the true value of the mean proliferation rate was within the 95% CI, this was achieved because the CIs were so wide. As wide CIs could be influenced by the width of the prior distribution, we repeated the fitting procedure using uniform prior distributions of different widths for $p$ (or $p_i$ for the explicit model). First, we reproduced the scenario proposed by Westera et al. [23], fitting implicit and explicit models to data generated using the explicit model and model parameters $p_1 = 0.72$, $p_2 = 0.016$, $\alpha_1 = 0.1$, using uniform priors [0,1] or [0,10] for $p_i$ (Fig 10A–10B). Note that to match the methods of Westera et al. [23], for Fig 10A–10B we assumed perfect labelling i.e. $U(t) = 1$ for $0 \leq t \leq \tau$ and $U(t) = 0$ otherwise. Upon widening the prior, the accuracy of the explicit kinetic heterogeneity models broke down resulting in very inaccurate estimates and extremely wide credible intervals. We then explored a wider range of parameters by fitting the homogeneous, implicit kinetic heterogeneity and two-compartment models to the previously simulated 125 data sets. Fig 10C shows that the errors in the estimates for the mean proliferation rate are prior-independent for the homogeneous and implicit kinetic heterogeneity models, but grow with the prior width for the two-compartment explicit heterogeneity model. The widths of the 95% CIs also grow with the prior width for the two-compartment model, approaching those of the prior (Fig 10D). This indicates that the estimates of mean proliferation rate obtained using the explicit heterogeneity models are heavily dependent on prior assumptions rather than the data, consistent with the high correlations between model parameters. Given that CFSE dilution studies of T cells, NK cells and B cells, both *in vitro* and *in vivo*, routinely report small subpopulations of cells undergoing large numbers of divisions (e.g. ≥ 6 divisions in 3 days, where 6 divisions represents the limit of detection in this case) it is hard to justify a conservative prior [24–26].

**Model selection does not improve parameter estimates.** We have shown that the implicit kinetic heterogeneity model and the explicit two-compartment model each produce good fits for different regions in parameter space, but we do not know to which region of parameter space a particular dataset corresponds. Also, if we choose to use an explicit model, we do not know the number of kinetically distinct subpopulations. One interesting approach, suggested by de Boer et al. [22], is to increase the number of compartments in the fitted explicit kinetic heterogeneity model until the goodness of fit is no longer improved. We investigate the effectiveness of this approach.

For the 125 simulated datasets above, we fit an $n$-compartment explicit kinetic heterogeneity model to the data (with a prior width of 1 for $p_i$). Model selection is implemented by first considering the two simplest models (with the fewest compartments). We use $\Delta elpd_{loo} = elpd_{loo,i} - elpd_{loo,j}$, which is a metric related to the expected log pointwise predictive density, as the criteria to compare these two models (Methods). Where there is insufficient evidence to select the more complex model, the simpler model is selected; where there is evidence to select the more complex model, the next most complicated model is added to the set of models under consideration, and model comparison is repeated. This process is repeated until the most complex model within the set of models under consideration is no longer favoured statistically, leading to selection of the second most complex model in the set.

Fig 8 compares the error in the model selection estimate, the width of the 95% CI and the percentage of accurate estimates of the mean proliferation rate versus using either the homogeneous, implicit heterogeneous or the two-compartment model as previously discussed.

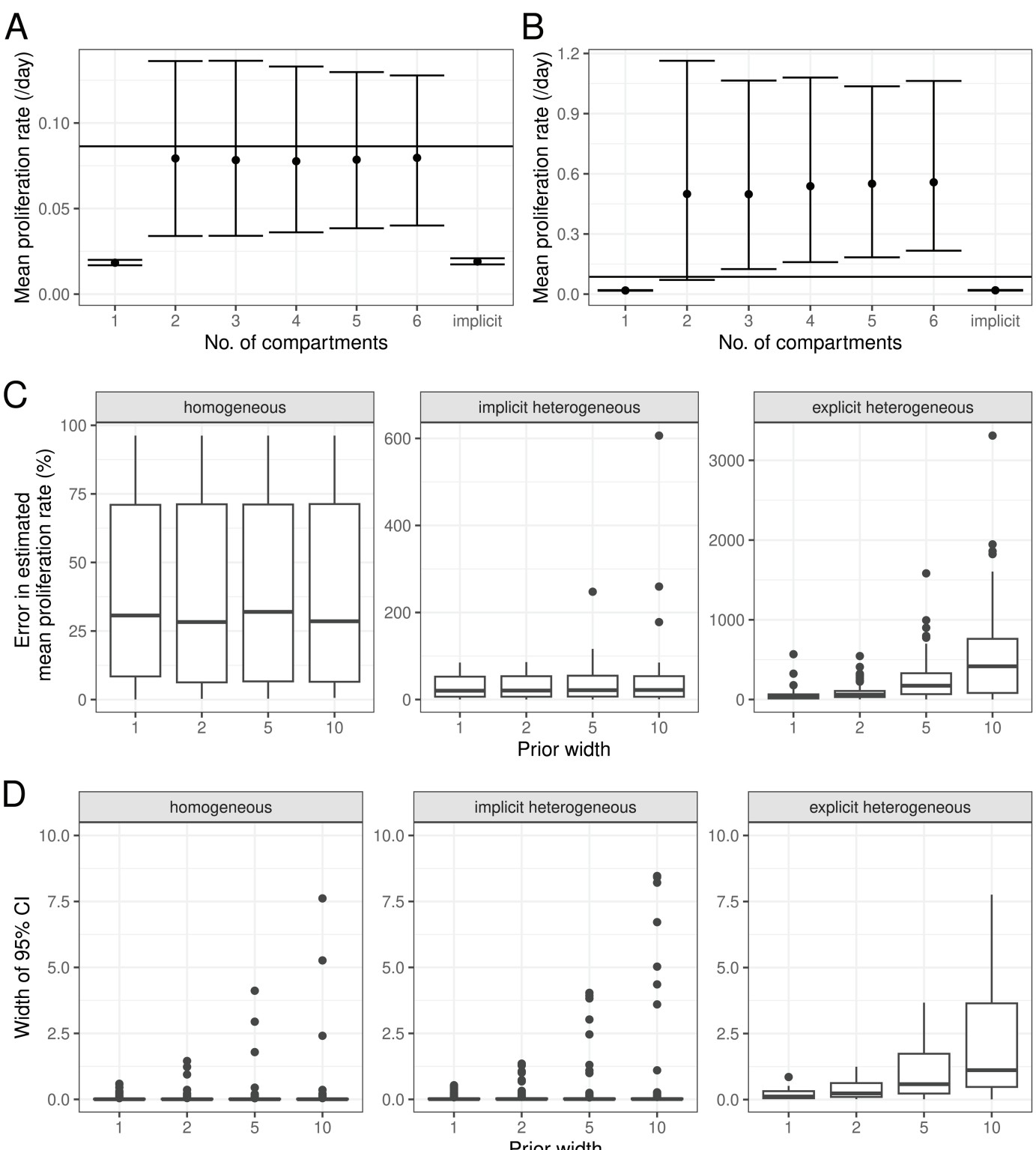

**Fig 10. Dependence of parameter estimates on the prior distribution.** (A–B): Estimated mean proliferation rates when fitting models with different priors to data generated using a two-compartment explicit heterogeneity model. Error bars indicate 95% credible intervals. On the $x$-axis, 'implicit' is the fit using the implicit kinetic heterogeneity model, while the others are using explicit models with the number of compartments indicated. (A) uses a uniform prior distribution across [0,1] day$^{-1}$ for $p_i$ (explicit model), $p$ and $d^*$ (implicit model), and (B) uses a uniform prior distribution across [0,10] day$^{-1}$ for these parameters. The horizontal line indicates the value used to simulate the data. (C–D): (C) The percentage error in the mean proliferation rate, and (D) the width of the 95% CI of the mean proliferation rate, across 125 simulations of the two-compartment model, when the models in the subplot titles are fitted to the data, with uniform priors for $p$ (or $p_i$) with upper bounds as indicated on the x-axis.

Model selection performs very similarly to the explicit two-compartment model in both its advantages and disadvantages. However, S13 Fig shows that the homogeneous model tends to be selected when the degree of heterogeneity is small, so this approach may lend additional evidence to the degree of heterogeneity. Results may vary if the stringency of the model selection is changed. We also note that as explicit heterogeneous models are considered, parameter estimates using the model selection method is likely be sensitive to the choice of prior distribution and that repeating the above analysis with less informative priors would results in degradation of the performance of the explicit two compartment model and the model selection approach.

We thus recommend choosing a model based on prior data rather than through model selection. If prior data is limited or uninformative then the implicit model should be used, but we should be aware that this model assumes a small degree of heterogeneity; greater heterogeneity may be present but undetectable in the data. On the other hand, if an informative prior can be generated, an explicit *n*-compartment model may be better. Where possible, we should constrain parameter space using additional data, such as telomere length [17,27] or Ki-67 expression data [28].

## 3. Spatial distribution of T lymphocytes

The final assumption we consider is the assumption of spatial homogeneity. Studies of T cell dynamics, at least for humans, are most often based on samples from the blood. Directly *ex vivo*, most peripheral blood T lymphocytes are in G0 \G1 [29,30]. Nevertheless high levels of deuterium are typically detected in peripheral blood lymphocytes following a labelling period. This indicates that division occurs outside the blood, most likely in lymphoid tissue, and that dividing cells take up label outside the blood stream and then travel through the blood where they are sampled. Here we ask whether cell proliferation and disappearance rates, estimated from the timecourse of labelled peripheral blood cells are accurate estimates of proliferation and disappearance of cells in lymphoid tissue. That is, do labelled cells in the blood provide a window into cell dynamics in lymphoid tissue? We address this question using the model represented by Eqs 8 and depicted in Fig 11.

$$\frac{dA}{dt} = pA - dA - gA + fB \tag{8a}$$

$$\frac{dB}{dt} = gA - fB. \tag{8b}$$

In which $A$ is the number of cells (of the cell population of interest) in lymphoid tissue and $B$ the number of cells of the same population in the blood. Cells in lymphoid tissue proliferate at rate $p$, disappear (die/ differentiate/ exit the circulation long-term) at rate $d$, enter the blood at rate $g$ and leave the blood at rate $f$. Upon labelling, then the fraction of label in each compartment ($F_A$ and $F_B$ respectively) is

$$\frac{dF_A}{dt} = pb_w U(t) - (d + g)F_A + fF_B \frac{\bar{B}}{\bar{A}}, \tag{9a}$$

$$\frac{dF_B}{dt} = gF_A \frac{\bar{A}}{\bar{B}} - fF_B \tag{9b}$$

where $\bar{A}$ and $\bar{B}$ are the steady state sizes of the populations $A$ and $B$ (i.e. $\bar{B}/\bar{A}$ is the blood to lymph ratio for the population of interest, which is of the order of 2/98 for lymphocytes [31])

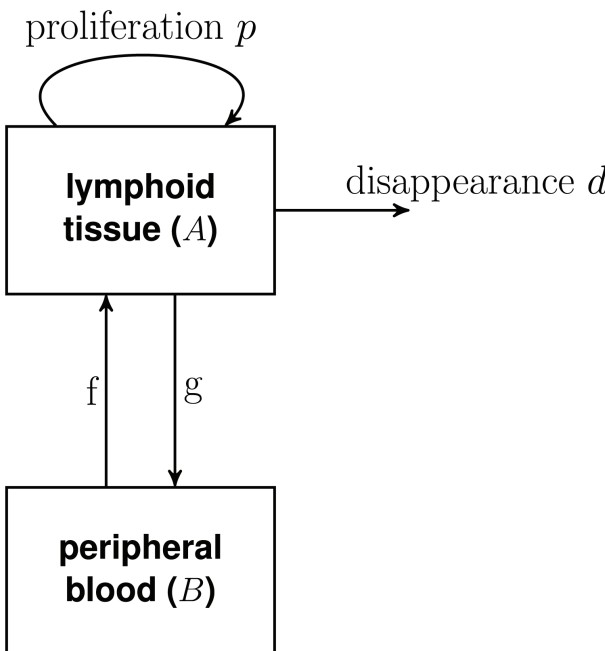

**Fig 11. Flow diagram for model with lymphoid tissue and blood.**

and $U(t)$ is the fraction of deuterium in body water (Eq 3). We assume that only label in the blood, $F_B$ can be observed.

We simulate this system using parameters from physiological ranges (Methods) and calculate the discrepancy between the label in the lymph and the observed label in the blood $D = max\left[\frac{F_A - F_B}{F_A}\right]$ for a range of values of $f$ and $p$. The maximum discrepancy is plotted in Fig 12 and a representative labelling plot for a particular set of parameters in Fig 13.

It can be seen that, for our best estimate of the lymphocyte recirculation rate ($f$=28 per day) then for all values of proliferation considered, the maximum discrepancy at any timepoint is extremely small. Even if the lymphocyte recirculation rate is ten-fold slower ($f$=0.28 per day), and the proliferation rate is high ($p$=0.1 per day) then the maximum discrepancy is still less than 2%. We conclude that, for freely recirculating T lymphocytes, then dynamics quantified from the blood will be representative of dynamics in lymphoid tissue.

## Discussion

We have investigated the impact of three widely used simplifying assumptions when modelling stable isotope labelling data. The first is the assumption that the target subpopulation being studied is closed. We considered the situation that there is an upstream precursor population flowing into the target cell population but that the upstream compartment is not sampled. When fitting a one compartment model to labelling data from the downstream compartment, label from any source, be it proliferation of the target population, proliferation of the precursor or clonal division upon differentiation is captured by the single parameter $p$. The correct interpretation of this parameter is therefore label that has arisen by division which we refer to as the production by division. $\hat{p}$ is therefore in all cases an accurate estimate of the production by division. In the past, $\hat{p}$ has been incorrectly interpreted as the proliferation of the target population. The total label arising from division is the sum of the label arising

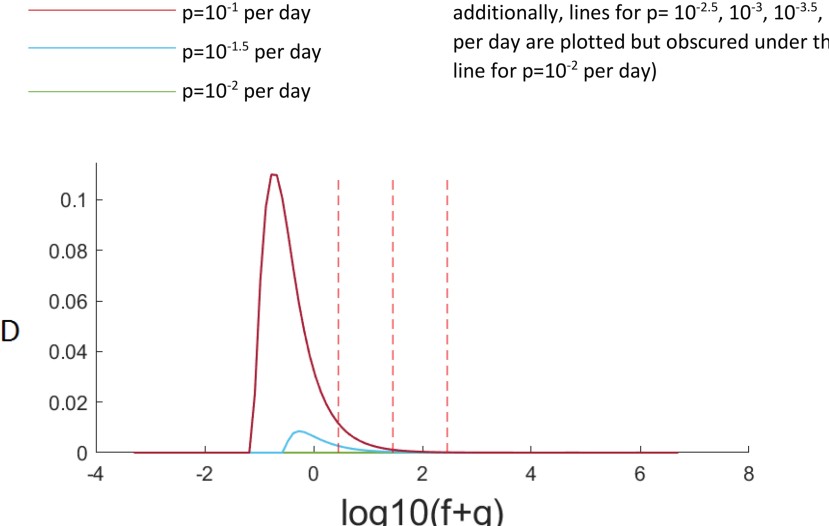

**Fig 12. Maximum discrepancy, D, between label in blood and label in secondary lymphoid tissue for varying lymphocyte proliferation rates (different coloured lines), varying $f + g$ along the x axis.** The central vertical dashed line marks the physiological recirculation rate of f=28 per day, lines to left and right mark 10 fold lower and higher (2.8 per day and 280 per day) respectively. It can be seen that unless recirculation is unrealistically slow and the lymphocyte kinetics are very fast then the label in the blood is a very good approximation of label in the lymphoid tissue.

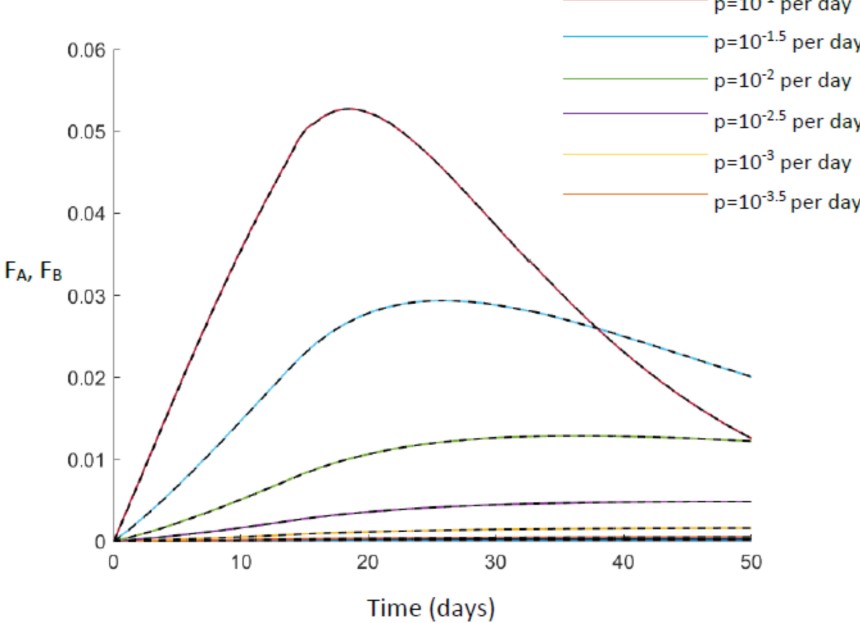

**Fig 13. A comparison of label in lymph node (solid line, $F_A$) and label in blood (dashed line, $F_B$).** It can be seen that, for all proliferation rates considered, they overlay. Here $f$=28 per day.

from the target population proliferation and the label arising from the upstream population; therefore, in the case where there is little label from the upstream population ($k = 0$ and small influx ratio), then the production by division is approximately equal to the proliferation and $\hat{p}$ approximates the proliferation rate. This approximation breaks down as the amount of label from the upstream population increases. On the other hand, turnover is equal to the gain (or loss) of all cells by the target population. Flow of unlabelled cells from the upstream population will be invisible in the labelling experiment, so if this flow is relatively large ($k = 0$ and large influx ratio), then turnover will be underestimated by the production by division. However, if this term is small ($k \gg 0$ or small influx ratio) then production by division will be similar to the turnover and therefore turnover will be accurately approximated by $\hat{p}$. As we usually do not know $k$ or the influx ratio, we recommend that the estimated parameter $\hat{p}$ be only interpreted as the production rate by division. Attempts to directly estimate proliferation by fitting the precursor/target model may suffer from large errors.

Swain et al. [35] addressed the same question but restricted to kinetically homogeneous upstream/target compartments, and for the special cases of $k = 0$ and $k = 1$, using different analytic derivations. As the kinetically homogeneous $k = 0$ and $k = 1$ cases are special cases of our model (heterogeneous $k \in [0, 20]$), the results of our two studies should agree for this case. For $k = 0$, we agree that the turnover rate may not be well estimated if the target compartment is assumed to be closed. Swain et al. [35] agrees with us that the turnover rate is well estimated when the turnover rate of the upstream compartment is fast, or when the inflow into the target compartment is small (i.e. there are few unlabelled cells from the precursor compartment).

The second assumption we analysed is that all lymphocytes in a population have the same turnover rate. Although previous studies have used implicit or explicit models to address kinetic heterogeneity, the circumstances under which each model should be used, and their limitations, have not been investigated.

We found that, even though the explicit model is the generative model then, comparing the estimates of proliferation obtained by fitting the implicit and explicit models, the implicit model had significantly smaller errors across the parameter range (P = 0.0047, Fig 8). The proportion of estimates falling in the CI was higher for the explicit heterogeneity model though at the cost of very wide CI. The only region of parameter space where the explicit heterogeneity model produced estimates with lower error was when there was considerable heterogeneity in the population ($p_1 \gg p_2$ and $\alpha$ small; Fig 9). Perhaps more worryingly, the estimates from the explicit heterogeneity model were heavily dependent on the prior, this was not seen for the implicit heterogeneity or homogeneous model (Fig 10). This indicates that the performance of the explicit heterogeneity model just discussed could degrade even further if we hadn't been using a prior which was closely aligned with the generating values. Additionally, we investigated whether a model selection approach was useful in the case when the number of subcompartments was unknown. We found little evidence that this approach yielded more accurate estimates of the mean proliferation and was liable to be dependent on the prior assumptions. We recommend that if prior data is limited or uninformative then the implicit heterogeneity model should be used; though this can lead to errors for populations with a high degree of heterogeneity. If an informative prior can be constructed then an explicit heterogeneity model may be preferable.

The third assumption is the assumption that lymphocytes are spatially homogeneous and in particular the assumption that the fraction of labelled lymphocytes quantified in the blood is representative of the fraction of labelled lymphocytes in the secondary lymphoid tissue. We found that this assumption was reasonable for lymphocytes that recirculated between blood

and lymphoid tissue at a rate of about 0.3 per day or faster. This is slow compared to the average rate of lymphocyte recirculation, so, provided a subpopulation is not recirculating much slower than average its kinetics should be reliably quantified from blood samples. If a lymphocyte population is sequestered in lymphoid tissue or only recirculating very slowly then label in the blood will be poorly representative of label in the secondary lymphoid tissue.

In section 1 (the upstream/downstream compartment section), we avoid the issue of the transient identity of cells undergoing clonal expansion (i.e. do they contribute to label as phenotype C or phenotype E cells) by assuming clonal expansion occurs in the lymph node and that by the time we sample in the blood the cells can be classified as belonging to the downstream compartment, i.e. phenotype switching is complete. Thus, in this case the spatial heterogeneity in the system is to our advantage.

In section 2 (the kinetic heterogeneity subsection) we restricted our study of heterogeneity to unconnected sub populations. It is possible to conceive of more complicated substructures in which there is flow between the subpopulations. The large number of possible models puts a wider investigation outside the scope of this manuscript but we suspect that, the conclusions will be broadly similar to what we have found with independent subpopulations, perhaps with a greater tendency towards favouring the implicit kinetic heterogeneity model as the flow will tend to reduce heterogeneity in the population.

More broadly, although we have discussed the processes of upstream and downstream compartments, kinetic heterogeneity and spatial heterogeneity separately, they may occur in tandem for a given situation. As there are many combinations of potential biases, we would first consider evidence for each of them separately. Given the complexity of the situation this evidence would need to be something other than the labelling data itself i.e. knowledge of the biological system.

In summary, we find that some frequently used assumptions can have profound impact on the interpretation and estimates of parameter values. Whilst these assumptions are typically unavoidable it is important to be aware of their implications when conducting labelling studies.

## Methods

Code to reproduce results can be found at https://github.com/ada-w-yan/cellkineticmodels.

### Simulating data for precursor/target model (Section *Upstream and downstream compartments*)

Transformed versions of the upstream and downstream parameters were sampled from a multivariate uniform distribution using Latin hypercube sampling. Table 1 shows the bounds of the distribution. Bounds above the line in Table 1 are bounds that were chosen to reflect physiological estimates. They include the bounds on $p_C$ and $p_E$ which were chosen in line with typical estimates of T cell proliferation in healthy individuals [32]. $0.01p_C \leq r \leq p_C$ was chosen so that the differentiation rate of the upstream population is smaller than its proliferation rate, as would be expected in homeostasis (the lower bound is imposed so there is flow between the two compartments; otherwise the problem reduces to the trivial solution of two independent compartments). The number of rounds of clonal expansion $k$ was initially chosen to lie between 0 and 20. 0 represents the scenario when cells flowing from C to E are unlabelled.

Bounds below the line are necessary to reflect model constraints. $1 \leq d_C^*/(p_C - r) \leq 100$ enforces the condition that $d_C \leq d_C^* \leq 100d_C$; i.e. the upstream population C has the freedom to be kinetically heterogeneous (with the maximum heterogeneity constrained such that the death rate of labelled cells is not greater than 100 times the average death rate). $0 \leq (\log_{10} d_E \,-$

**Table 1. The bounds of the distribution from which transformed parameters are sampled.** All rates are in units day$^{-1}$.

| Parameter | Bounds |
|---|---|
| $\log_{10} p_C$ | [−4,−2] |
| $\log_{10} p_E$ | [−4,−2] |
| $\log_{10}(r/p_C)$ | [−2,0] |
| $k$ | [0,20] |
| $\log_{10}(d_C^*/(p_C - r))$ | [0,2] |
| $(\log_{10} d_E - \log_{10} p_E)/(\log_{10} \delta - \log_{10} p_E)$ | [0,1] |
| $(\log_{10} d_E^* - \log_{10} p_E)/(\log_{10} \delta - \log_{10} p_E)$ | [0,1] |

$log_{10} p_E)/(\log_{10} \delta - \log_{10} p_E) \leq 1$ and $0 \leq (\log_{10} d_E^* - log_{10} p_E)/(\log_{10} \delta - \log_{10} p_E) \leq 1$ impose $p_E \leq d_E \leq \delta$ and $p_E \leq d_E^* \leq \delta$ respectively. The lower bounds are necessary for the target population to remain in equilibrium and the upper bounds reflect the design of stable isotope labelling experiments which mandates that the loss of the isotope from the body ($\delta$) must be more rapid than the loss of the population of interest. In addition, we only keep samples that fulfil $d_E^* > d_E$. This allows the target population to be kinetically heterogeneous. The value of $\delta$ was fixed to 0.07 which is a realistic value from the literature [2].

S1 Fig shows the shape of the distribution in terms of the original parameters.

We generate 100 parameter samples from the above distribution and 100 parameter samples from the version with $k = 0$. For each sample, we substituted parameter values into the analytic solution of Eq 2 assuming a labelling period of 49 days. We take data points daily up to day 100.

**Optimal data.** Initially we simulated an optimal experimental situation with a large number of points with a small standard deviation because the aim is to determine the effect of the upstream compartment and not the effect of uncertainty in the data. We take data points daily. Each data point is lognormally distributed with parameters $\mu = \log(x)$ where $x$ is the true value of the fraction of label, and $\sigma = 0.005$.

**Realistic data.** Although our focus was on "optimal data" we also investigated how our conclusions varied with more realistic data. The realistic data differed from the optimal data in three main respects. First, data sampling was assumed to be much sparser: we assumed measurements once every week, resulting in 15 measurements over the 100 day period of the experiment. Second, we allowed for considerably more noise: noise was normally distributed with mean zero and standard deviation 0.1× max value of label in that data set. Finally, we explicitly modelled heterogeneity in the target compartment. That is, we modelled two subpopulations within the target compartment (each independently in equilibrium) rather than a single population with $d^* > p$. The equations to describe label in this model are as follows:

$$\frac{dF_C}{dt} = p_C b_w U(t) - (d_C^* + r)F_C, \tag{10a}$$

$$\frac{dF_{E1}}{dt} = \gamma(2^k - 1)b_w U(t)r\frac{\bar{C}}{\bar{E}_1} + \gamma r\frac{\bar{C}}{\bar{E}_1}F_C + p_1 b_w U(t) - d_1 F_{E1} \tag{10b}$$

$$\frac{dF_{E2}}{dt} = (1-\gamma)(2^k - 1)b_w U(t)r\frac{\bar{C}}{\bar{E}_2} + (1-\gamma)r\frac{\bar{C}}{\bar{E}_2}F_C + p_2 b_w U(t) - d_2 F_{E2} \tag{10c}$$

Where $F_C$ is the fraction of label in the upstream compartment and $F_{E1}$ and $F_{E2}$ is the fraction of label in the two target subpopulations. Cells in the upstream compartment proliferate

**Table 2. Parameter values used to simulate the kinetic heterogeneity data.** All rates have units day$^{-1}$.

| Parameter | A, D | B, E | C, F |
|---|---|---|---|
| $p_1$ | varied | 0.018 | 0.72 |
| $p_2$ | 0.016 | varied | 0.016 |
| $\alpha_1$ | 0.7 | 0.1 | varied |
| $f$ | 0.032 | | |
| $\delta$ | 0.064 | | |
| $b_w$ | 4.18 | | |

at rate $p_C$, are lost at rate $d_C^*$ and differentiate at a rate $r$. Upon differentiation, cells undergo $k$ rounds of clonal division and a fraction $\gamma$ enter target subpopulation $E_1$ (with the remainder, $1 - \gamma$, entering subpopulation $E_2$). Cells in the two subpopulation proliferate at rates $p_1$ and $p_2$ respectively and are lost at rates $d_1$ and $d_2$ respectively. The steady state sizes of the three populations are denoted $\bar{C}$, $\bar{E}_1$ and $\bar{E}_2$. $U(t)$ is the availability of label in the body water and $b_w$ the normalisation factor as in Eq 2.

### Fitting simulated data (Section *Upstream and downstream compartments*)

When fitting the one compartment model alone to the simulated data then rstan version 2.19.2 was used. Code is written in R version 3.6.3. The prior distributions for $p_E$ and $d_E^*$ were uniform between 0 and $5/b_w$, and between 0 and 1 respectively.

When fitting the two compartment precursor/target model we switched to a frequentist approach due to convergence problems with stan. This was implemented via the Pseudo function in the R package FME. In order to ensure fits were comparable, whenever performing a direct comparison with the two compartment model (i.e. Section *Comparing fits using models with and without an upstream compartment*) then the one compartment model was also fitted using FME. The 95% confidence intervals of the parameter estimates were found by bootstrapping the data. That is, 500 bootstrap data sets the size of the original data set were created by randomly sampling with replacement from the original data set. Each bootstrap data set was fitted and the parameters estimated. The 95% CI were taken to be the 0.025 and 0.975 percentiles of the 500 parameter estimates.

### Simulating data for two-compartment explicit heterogeneity model (Section *Kinetic heterogeneity*)

Data were simulated using Eq 6 with $N$ = 2, and all combinations of $p_1 = 0.0072, 0.018, 0.036, 0.12, 0.24, 0.36, 0.72$; $p_2 = 0.0036, 0.0072, 0.0108, 0.0160$; $\alpha_1 = 0.1, 0.3, 0.5, 0.7, 0.9$ where $p_1 > p_2$ (that is a total of 125 parameter combinations). The other parameters were fixed at $\delta$ = 0.07, $f$ = 0.032, $b_w$ = 4.18 and the labelling period is 49 days. Errors on simulated data were lognormally distributed with $\sigma$ = 0.005. Data points were taken every 7 days up to 98 days.

### Fitting data for two-compartment kinetic heterogeneity model (Section *Kinetic heterogeneity*)

The data were fitted using the No U-Turn sampler implemented in the rstan package. Three models were considered: the explicit heterogeneity model (Eq 6), the implicit heterogeneity model (Eq 7) and a homogeneous model (i.e. $p = d^*$ in Eq 7). The body water parameters $f$ and $\delta$, the amplification factor $b_w$, and the standard deviation for the sampling error, were assumed to be known.

The homogeneous model only has one fitted parameter, $p$, and its prior distribution was uniform between 0 and 1.

The explicit heterogeneity model has $2N–1$ fitted parameters, where $N$ is the number of compartments. The prior distributions for the proliferation rates $p_i$ are independent; each has a uniform prior of [0,1]. The prior distribution for $\alpha_i$ is symmetric Dirichlet with concentration parameter $\alpha = 1$. This prior distribution enforces $\sum_{i=1}^{N} \alpha_i = 1$. Without loss of generality we order the vector $\alpha_i$ from smallest to largest. For example, for $N = 2$ this is equivalent to a uniform prior for $\alpha_1$ between 0 and 0.5, and setting $\alpha_2 = 1 – \alpha_1$. The implicit heterogeneity model has two fitted parameters, $p$ and $d^*$. The prior distribution is independent between parameters, with a uniform prior of [0,1] for $p$ and $d^*$. The one exception to these priors is in Fig 10 where we explored the impact of prior assumptions. Here the prior distribution for $p$ in the homogeneous model is [0,$x$] where $x = 2, 5$, or 10. For the implicit heterogeneity model, priors are [0,$x$] for $p$ and $d^*$. For the explicit heterogeneity model with $N = 2$, priors are [0,$x$] for $p_1$ and $p_2$ (the prior for $\alpha_1$ is unchanged at [0,0.5]).

## Model comparison (Section *Kinetic heterogeneity*)

We compare the fits between models with $i$ and $j$ compartments by computing the point estimate and standard error for $\Delta elpd_{loo} = elpd_{loo,i} – elpd_{loo,j}$ using the *loo* package version 2.4.1 in R. $elpd_i$ is the expected log pointwise predictive density for a model with $i$ compartments, and $elpd_{loo,i}$ is its estimate using leave-one-out cross-validation [33]. We define support for model $i$ over model $j$ when the point estimate for $\Delta elpd_{loo}$ is positive and exceeds its standard error. We define that two models are tied if the standard error for $\Delta elpd_{loo}$ is greater than the magnitude of its point estimate.

## Calculating the discrepancy between label in the blood and in the lymphoid tissue (Section *Spatial distribution of T lymphocytes*)

The discrepancy between label in blood and label in lymphoid tissue was calculated for parameters in the ranges in Table 3.

We set $\bar{B}/\bar{A}$, the blood to lymph ratio of interest, to 2/98. Bounds on $p$ were chosen in line with typical estimates of T cell proliferation in healthy individuals [32]. Our baseline estimate of $f$ is calculated from the observation that there are $10^{10}$ lymphocytes in the blood and that every day $2.5 \times 10^{11}$ lymphocytes exit blood for the spleen and $0.3 \times 10^{11}$ lymphocytes per day exit the blood for the lymph node; giving an estimate of $f = 2.8 \times 10^{11}/10^{10} = 28 d^{-1}$. Estimates taken from Kuby Immunology [34] are a similar order of magnitude: there it is reported that it takes 30 mins for a lymphocyte to transit the blood compartment and that 87% of lymphocytes exit for lymphoid tissue giving $f = 42 d^{-1}$. These estimates of $f$ are then varied over several orders of magnitude ($log_{10}f \in [-4, 8]$) to ensure that results are robust. Given a value of $p$ then $d$ is constrained by the equilibrium constraint $\dot{A} = 0$ which requires $d = p$ and so the bounds on $d$ are the same as the bounds on $p$ but only $p$ is independently sampled. Similarly,

**Table 3. The parameter ranges for calculating the discrepancy between label in blood and label in lymphoid tissue.** All rates are in units day$^{-1}$.

| Parameter | Bounds |
|---|---|
| $log_{10} p$ | $[-4,-1]$ |
| $log_{10} f$ | $[-4, +8]$ |

the equilibrium constraint $\dot{B} = 0$ together with the assumption that approximately 2% of lymphocytes are in blood means that $g = f\bar{B}/\bar{A}$.

## Supporting information

**S1 Fig. Distribution from which simulation parameters are sampled for the Upstream and Downstream Compartments section.**
(PDF)

**S2 Fig. The relationship between model parameters and error in $\hat{d}_E^*$.** Top: the ratio between $r\frac{\bar{C}}{\bar{E}}$ and the production rate by division, for (A) $1 \leq k \leq 20$ and (B) $k = 0$. Bottom: the ratio between $p_C \frac{\bar{C}}{\bar{E}}$ and the production rate by division, for (C) $1 \leq k \leq 20$ and (D) $k = 0$.
(PNG)

**S3 Fig. Error in estimates of the proliferation rate.** This shows the same data as in Fig 6A but without the y axis truncation.
(PDF)

**S4 Fig. Error in estimates of the proliferation rate for realistic data but with a high proliferation rate. A** Discrepancy between point estimate of the proliferation rate and the true value (expressed as a fraction of the true value) for the one compartment model (red), precursor/target model (blue) and precursor/target with ratio model (green).On increasing the values of $p_E$ used to generate the labelling data two effects were noticed compared to the results depicted in Fig 6: first, errors were now much lower, second, there was no longer any significant difference in the size of the errors associated with each of the three models. P values are calculated by Wilcoxon signed rank, two-tailed; not corrected for multiple comparisons (number of independent comparisons $\leq 3$). **B** 95% confidence intervals (CI) were estimated by bootstrapping the data (Methods) and the fraction of runs where the true value lay within the CI was reported. Colours as for A. Compared to the corresponding figure for realistic data but with a lower value of $p_E$ (Fig 6) the proportion of runs where the estimate fell within the CI was still very low but the pattern across the models was quite distinct, for this set of parameters the one compartment model outperformed the other two models.
(JPG)

**S5 Fig. Relationship between $\hat{p}_E$ and the target population descriptors.** Estimated $\hat{p}_E$ compared to (from left to right) the rates of proliferation, turnover and production by division of cells in the target compartment. Top: for the case when fitting the one compartment model; middle: for the case when fitting the precursor/target model bottom: for the case when fitting the precursor/target model with ratio.
(PDF)

**S6 Fig. Median and 95% credible intervals for the mean proliferation rate when fitting the implicit model to data generated using the two-compartment explicit model.** The value of $p_1$ used to simulate the data is shown on the x-axis. Within each plot, the values of $p_2$ and $\alpha_1$ used to simulate the data are held constant at the values on the top and right respectively.
(PDF)

**S7 Fig. Median and 95% credible intervals for the mean proliferation rate when fitting the implicit model to data generated using the two-compartment explicit model.** The value of

$\alpha_1$ used to simulate the data is shown on the x-axis. Within each plot, the values of $\alpha_1$ and $p_1$ used to simulate the data are held constant at the values on the top and right respectively. (PDF)

**S8 Fig. Median and 95% credible intervals for the mean proliferation rate when fitting the implicit model to data generated using the two-compartment explicit model.** The value of $\alpha_1$ used to simulate the data is shown on the x-axis. Within each plot, the values of $p_2$ and $p_1$ used to simulate the data are held constant at the values on the top and right respectively. (PDF)

**S9 Fig. Median and 95% credible intervals for the mean proliferation rate when fitting the two-compartment explicit model to data generated using the same model.** The value of $p_1$ used to simulate the data is shown on the x-axis. Within each plot, the values of $p_2$ and $\alpha_1$ used to simulate the data are held constant at the values on the top and right respectively. (PDF)

**S10 Fig. Median and 95% credible intervals for the mean proliferation rate when fitting the two-compartment explicit model to data generated using the same model.** Median and 95% credible intervals for the mean proliferation rate when fitting the the two-compartment explicit model to data generated using the same model. The value of $\alpha_1$ used to simulate the data is shown on the x-axis. Within each plot, the values of $\alpha_1$ and $p_1$ used to simulate the data are held constant at the values on the top and right respectively. (PDF)

**S11 Fig. Median and 95% credible intervals for the mean proliferation rate when fitting the two-compartment explicit model to data generated using the same model.** Median and 95% credible intervals for the mean proliferation rate when fitting the the two-compartment explicit model to data generated using the same model. The value of $\alpha_1$ used to simulate the data is shown on the x-axis. Within each plot, the values of $p_2$ and $p_1$ used to simulate the data are held constant at the values on the top and right respectively. (PDF)

**S12 Fig. Pairs plots of the posterior distributions for fitting the two-compartment model to data generated with two different sets of parameter values.** (PDF)

**S13 Fig. The model which is selected for each combination of true parameter values, when performing model selection for kinetic heterogeneity.** (PDF)

**S1 File. Derivation of approximations for upstream and downstream compartment model.** (PDF)

**S2 File. Parameters for simulation of upstream and downstream compartment model.** (XLSX)

**S3 File. Simulated datasets for upstream and downstream compartment model.** (PDF)

## Acknowledgments

We are very grateful to Arpit Swain, José Borghans and Rob de Boer for helpful discussions.

## Author contributions

**Conceptualization:** Ada Wing Chi Yan, Ildar Sadreev, Yan Zhang, Derek Macallan, Robert Busch, Becca Asquith.

**Formal analysis:** Ada Wing Chi Yan, Ildar Sadreev, Becca Asquith.

**Funding acquisition:** Becca Asquith.

**Investigation:** Ada Wing Chi Yan, Ildar Sadreev, Jonas Mackerodt, Becca Asquith.

**Methodology:** Ada Wing Chi Yan, Ildar Sadreev, Jonas Mackerodt, Becca Asquith.

**Project administration:** Becca Asquith.

**Software:** Ada Wing Chi Yan, Becca Asquith.

**Supervision:** Becca Asquith.

**Visualization:** Ada Wing Chi Yan, Ildar Sadreev, Becca Asquith.

**Writing – original draft:** Ada Wing Chi Yan, Ildar Sadreev, Becca Asquith.

**Writing – review & editing:** Ada Wing Chi Yan, Ildar Sadreev, Yan Zhang, Derek Macallan, Robert Busch, Becca Asquith.

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
