## [Decision Letter · Decision Letter 0]

5 Aug 2024

Dear Dr. Asquith,

Thank you very much for submitting your manuscript "The Impact of Model Assumptions in Interpreting Cell Kinetic Studies" for consideration at PLOS Computational Biology.

As with all papers reviewed by the journal, your manuscript was reviewed by members of the editorial board and by several independent reviewers. In light of the reviews (below this email), we would like to invite the resubmission of a significantly-revised version that takes into account the reviewers' comments.

As you will see the reviewers were generally positive and found your work sound and clearly presented. Reviewer 1 and 2 your work as a valuable contribution to the literature on immune cell turnover. Reviewer 1 highlights a few relevant studies that you could consider citing or even discussing. Reviewer 2 would appreciate more quantitative analysis of the bias arising from model misspecifications. Reviewer 3 found your study to lack novelty. We would therefore encourage you to work out more clearly how your study goes beyond previous studies, either formally or conceptually.

We cannot make any decision about publication until we have seen the revised manuscript and your response to the reviewers' comments. Your revised manuscript is also likely to be sent to reviewers for further evaluation.

Sincerely,

Roland R Regoes

Academic Editor

PLOS Computational Biology

Amber Smith

Section Editor

PLOS Computational Biology

Reviewer's Responses to Questions

**Comments to the Authors:**

Reviewer #1: This is a thorough study exploring the caveats in the interpretation of deuterium labelling experiments.

It is obviously targeted at our fairly small community of labelling enthusiasts (!) but it still presupposes a lot of knowledge of the basics. It would be good to give a more detailed intro to the concepts - inheritance of label by DNA strands, etc.

On the issue of where cells “go” between C and E. If E is fed by clonal expansion of C the transition is likely to be blurry - that is, cells may be classified as members of the target population while they are still dividing. If this is true it obviously becomes a bit of a mess - basically a KH model with a transitory fast-dividing state. This could be worth mentioning. This relates to the larger issue that in practice these processes you consider could be operating in tandem. Could you discuss how you might approach this issue?

Kinetic heterogeneity. The freedom here is assumed to be the number of independent compartments - but there are many possible relationships between kinetically distinct subsets - e.g. they might be related linearly, as a series of differentiated sub-states within the target population. Could you comment/speculate on whether this matters for your analysis?

Your Discussion summarises the results nicely. But I found the sections describing explorations of parameter spaces a little lengthy and hard to follow. I think the paper would read better if you condensed the numerical explorations and focused on more intuitive explanations of WHY you expect these biases to occur. I think it might also be informative to show correlations between parameters - so you can discuss identifiability as well as biases in parameter estimates.

it’s not 2H labelling but our study Lukas et al. Front Imm 2023 models the kinetics of division-linked labelling within the thymus and naive pool - our approach was to use an empirical description of the label freq kinetics within late-stage thymic precursors (your F_C (t)) to allow us to measure the turnover (loss rate and hence mean residence time) of naive T cells. For any given target population, we may have a few candidate precursor populations that can be treated in the same way, and fits to the target pop data with different precursors could be compared. We have also modelled dynamics of non-closed and kinetically heterogeneous target populations in several other studies (Gossel eLife 2017, Hogan eLife 2019, Bullock Plos Bio 2024, in press and currently on bioarxiv), which might be of interest.

Other minor points:

L82

2^k >>1 is the criterion for influx-dominated production. is it not more precise to say that it’s (2^k-1)b_w U(t) >> F_C ? In principle you could have a lot of clonal expansion (and hence label uptake) happening within the precursor but only a small fraction of it entering the target.

L82-85

You’re dealing with the 1 compartment model here, so perhaps d^*_E should be just d_E?. Or clarify here in advance that you are always fitting the p d-star model

L120

“The precursor cells turn over quickly” - now you use turn over in a different sense to your definition. Do you mean “the precursor cells proliferate extensively before entering the target compartment”?

L132 “Unsurprisingly” - needs the detail on why you expect parameter identifiability issues before you say this

L338 What does “accurate assessment of prior data” mean? it would be great to include a more explicit recipe for how to proceed.

L359 the number 2/98 needs a reference

Lastly - I found the terminology a little confusing. This is not a huge issue as it’s mostly self- consistent, but I think it could be clearer. I think it’s more reasonable to describe self-renewal of existing memory cells as “division”, since most models describe this with single divisions. “Proliferation” is maybe better reserved for multiple divisions (clonal expansion)? For example, your phrase “rate of production by division” seems more closely aligned to self-renewal than influx from clonally expanded cells. And L87 - “then the proliferation rate is equal to the production rate by division” - this sounds true by definition.

Turnover is defined in other contexts as loss, so I advocate for a different terminology. Not required but food for thought:

Turnover = loss = disappearance (death plus differentiation)

Production = influx + self-renewal

And breaking it down further, influx = proliferation-linked influx + passive influx (or something similar)

At steady state, production = turnover

p_E is then the rate of self-renewal, not the proliferation rate

An alternative (and probably more useful) proposal is that you align the terminology with the companion paper from Swain et al.!

Reviewer #2: In this manuscript, the authors address the important question how necessary model assumptions when analysing data of isotope-labelling studies might influence the inferred kinetics of cellular turnover. In particular, they investigate the impact of three model assumptions, i.e. that the cell population is (1) from a single compartment, (2) kinetic homogeneous, and (3) spatially homogeneous. To this end they use simulated data and test the inference of the used parameters by subsequently fitting the different models to the data.

The study represents a valuable contribution to the interpretation of cellular turnover dynamics using isotope labelling data indicating potential scenarios of misinterpretation of parameters. The manuscript is well written and the argumentation can be easily followed. However, I would have some comments that the authors might want to consider.

# Major points:

1.) In the analyses, the authors investigate to which extent the inference of cellular kinetics is impaired by inappropriate modelling assumptions for isotope labelling data. Most of the evaluations and comparisons are given in a qualitative way (i.e. "very similar", "much less", "a significant flow of labelled cells" etc) which partly makes it difficult to judge, under which conditions the inappropriate model assumptions clearly play a role., i.e., what exactly defines a significant flow. Given that they checked the robustness of the observations based on several parameter combinations (e.g. Figure 3) to show the generality of their observations, I was wondering if it would not be possible to make these statements a bit more quantitative, i.e., under which relationships of flux rates or loss rates you would expect which kind of deviation, which could help to assess potential correction factors.

2.) Equation 1 and line 49: I would strongly recommend to replace the symbol $\Delta$ (capital delta) for the differentiation rate as this will be easily misread as the Laplace-Operator within differential equations.

3.) Within the spatial model, the authors assume a recirculation of cells between lymphoid tissue and peripheral blood finding mostly an overlay between the dynamics of labelled cells within the two compartments (e.g. Figure 12), i.e. indicating a reasonable interpretation of measurements within this context. However, the model neglects any loss of cells from the blood (e.g. migration of cells to peripheral tissue), which would remove cells from the circulation and could also happen at a different rate than the disappearance of cells from the lymphoid tissue. I think this could be a relevant factor when considering physiological conditions and might be worth to discuss or investigate.

# Minor points:

- Within the figures the axis labelling and axis annotations are partly very difficult to read (e.g. Figure 4)

- Figure 4: Were the statistical tests used for comparison corrected for multiple comparisons?

Reviewer #3: In this manuscript, authors assess the impact of three model assumptions for fitting stable isotope labelling data. They use simulated data to investigate the impact of model assumptions on parameters estimations and the accuracy of these estimations. I found the paper easy to follow, however I think it could be streamlined by reducing some sections and better highlighting the new results. For me it's unclear what are the biological implications of this work and the impact in isotope labelling experiments. It does not seem that the authors come up with specific methological recommendations based on their results (except "balancing accuracy with precision", which is pretty vague).

Please find some detailed comments below:

- overall comment: could you make a table that summarises the questions asked / results obtained? In the intro it seems that you’ll only address 3 points but some are divided in further sub-questions and it would be nice to get an overview of this

- is paragraph lines 76-89 necessary? I find it very vague and approximate. Could it be reduced and used as quick introduction for the following paragraph?

- section starting line 90: could you comment (maybe in the discussion) on the error margins here and how much of an impact it makes in isotope labelling studies? (e.g., Fig 2 if the true value is 0.007 and the estimate around 0.002, what does it mean in terms of interpretation of results in a given labelling study, and what are the implications of such error?)

- Lines 211-224 is more “introduction” and not really results. I would reduce to focus more on the new results presented by this paper. It's also unclear what results are new in paragraph lines 211-240.

- Line 218: you mention "underestimation of mean proliferation rate". Fig D shows over-estimation of this rate, does that mean that your results are not in line with Westera et al?

- Figure 6: could you try to use same scales for the y-axis? I find the comparison A-D, B-E, C-F difficult to make visually

- Lines 255-256 "This indicates that the estimates of mean proliferation rate obtained using the explicit heterogeneity models are heavily dependent on prior assumptions": I think to state this you would need to check the results for more than just 2 priors

- Lines 270-271, 275, when you mention the "lower bound" and "upper bound". I am not convinced by these claims. How helpful is a lower/upper bound if we don't know how accurate this bound is? For example if I read Fig 8 properly, the proliferation rate estimation can be around 0.5 in some cases, while the parameter is 0.0036. I'm not sure this upper bound value is helpful in any way.

- Lines 278-289: is this more for the discussion?

- line 151 you mention 100 simulated datasets (maybe refer to the methods here, as there is no mention of any simulated datasets before in the results section), but line 301 you refer to 125 simulated datasets. I don't find any mention of 125 simulations in the methods. Could you please clarify?

- minor: some figures have titles, some not. Please harmonise

**Have the authors made all data and (if applicable) computational code underlying the findings in their manuscript fully available?**

Reviewer #1: Yes

Reviewer #2: Yes

Reviewer #3: Yes

PLOS authors have the option to publish the peer review history of their article (what does this mean?). If published, this will include your full peer review and any attached files.

Reviewer #1: No

Reviewer #2: No

Reviewer #3: No
---

## [Decision Letter · Decision Letter 1]

19 Nov 2024

PCOMPBIOL-D-24-00941R1The Impact of Model Assumptions in Interpreting Cell Kinetic StudiesPLOS Computational Biology  Dear Dr. Asquith, Thank you for submitting your manuscript to PLOS Computational Biology. The reviewers are largely very happy with your revisions. Reviewer #2 raises a last point about the coverage of parameter space in Fig 6A, which we would encourage you to consider. Please submit your revised manuscript within 30 days Jan 19 2025 11:59PM. If you will need more time than this to complete your revisions, please reply to this message or contact the journal office at ploscompbiol@plos.org. Please include the following items when submitting your revised manuscript: * A rebuttal letter that responds to each point raised by the editor and reviewer(s). You should upload this letter as a separate file labeled 'Response to Reviewers'. This file does not need to include responses to formatting updates and technical items listed in the 'Journal Requirements' section below. * A marked-up copy of your manuscript that highlights changes made to the original version. You should upload this as a separate file labeled 'Revised Manuscript with Track Changes'. * An unmarked version of your revised paper without tracked changes. You should upload this as a separate file labeled 'Manuscript'. If you would like to make changes to your financial disclosure, competing interests statement, or data availability statement, please make these updates within the submission form at the time of resubmission. Guidelines for resubmitting your figure files are available below the reviewer comments at the end of this letter.  We look forward to receiving your revised manuscript. Kind regards,Roland R RegoesAcademic EditorPLOS Computational Biology

Amber Smith

Section Editor

PLOS Computational Biology

Feilim Mac Gabhann

Editor-in-Chief

PLOS Computational Biology

Jason Papin

Editor-in-Chief

PLOS Computational Biology

**Journal Requirements:**

We have noticed that you have not cite Table 3 in the manuscript file. Please add in-text citation for all figures and tables within your manuscript.

**Reviewers' comments:** Reviewer's Responses to Questions

**Comments to the Authors:**

Reviewer #1: Thanks to the authors for the careful responses, and the extensive and clarifying revisions.

Reviewer #2: The authors have performed additional analyses to address the comments brought forward by the reviewers, substantially revising the manuscript. Although there seems to be an issue with the Figure numbering in response to my previous comments, the authors have addressed them appropriately. I see the difficulty of providing general relationships and correction factors to avoid misinterpretation of parameter estimates, which slightly limits the applicability of the study, but nevertheless raises the awareness of the misinterpretation problem and strategies to overcome them. I would only have one minor comment which concerns the statistical analyses, as e.g. performed in Figure 6:

If I understand correctly, the statistics and comparisons (as e.g. Figure 6A) are performed on 100 simulated data sets of the two compartment model using "a wide range of parameters" (line 131). Given that Eq.2 contains several free parameters, I think it could be relevant to check that the 100 parameter combinations truly cover a wide range for each of the parameters used, i.e. by showing the distribution. How many parameters were actually varied? It is not directly clear if 100 parameter combinations are sufficient to address a large range of parameter combinations. In addition, do these parameter combinations generally lead to "informative dynamics", e.g. not immediate loss of all cells, which might affect the ability to infer meaningful dynamics? This is only a minor point which could affect the comparisons, but I apologise if this has already been addressed and I missed this within the text.

**Have the authors made all data and (if applicable) computational code underlying the findings in their manuscript fully available?**

Reviewer #1: Yes

Reviewer #2: None

PLOS authors have the option to publish the peer review history of their article (what does this mean?). If published, this will include your full peer review and any attached files.

Reviewer #1: No

Reviewer #2: No

**Figure resubmission:** While revising your submission, please upload your figure files to the Preflight Analysis and Conversion Engine (PACE) digital diagnostic tool, https://pacev2.apexcovantage.com/. PACE helps ensure that figures meet PLOS requirements. To use PACE, you must first register as a user. Registration is free. Then, login and navigate to the UPLOAD tab, where you will find detailed instructions on how to use the tool. If you encounter any issues or have any questions when using PACE, please email PLOS at figures@plos.org. Please note that Supporting Information files do not need this step. If there are other versions of figure files still present in your submission file inventory at resubmission, please replace them with the PACE-processed versions.**Reproducibility:** To enhance the reproducibility of your results, we recommend that authors of applicable studies deposit laboratory protocols in protocols.io, where a protocol can be assigned its own identifier (DOI) such that it can be cited independently in the future. Additionally, PLOS ONE offers an option to publish peer-reviewed clinical study protocols. Read more information on sharing protocols at https://plos.org/protocols?utm_medium=editorial-email&utm_source=authorletters&utm_campaign=protocols

---

## [Editor Report · Decision Letter 2]

9 Dec 2024

Dear Dr. Asquith,

We are pleased to inform you that your manuscript 'The Impact of Model Assumptions in Interpreting Cell Kinetic Studies' has been provisionally accepted for publication in PLOS Computational Biology.

Best regards,

Roland R Regoes

Academic Editor

PLOS Computational Biology

Amber Smith

Section Editor

PLOS Computational Biology

Feilim Mac Gabhann

Editor-in-Chief

PLOS Computational Biology

Jason Papin

Editor-in-Chief

PLOS Computational Biology

---

## [Editor Report · Acceptance letter]

PCOMPBIOL-D-24-00941R2

The Impact of Model Assumptions in Interpreting Cell Kinetic Studies

Dear Dr Asquith,

I am pleased to inform you that your manuscript has been formally accepted for publication in PLOS Computational Biology. Your manuscript is now with our production department and you will be notified of the publication date in due course.

With kind regards,

Lilla Horvath
